# Flower Visitation Time and Number of Visitor Species Are Reduced by the Use of Agrochemicals in Coffee Home Gardens

Sophie Manson [1], K. A. I. Nekaris [1], Katherine Hedger [2], Michela Balestri [1], Nabil Ahmad [2], Esther Adinda [2], Budiadi Budiadi [3], Muhammad Ali Imron [3], Vincent Nijman [1] and Marco Campera [1,*]

1  School of Social Sciences, Oxford Brookes University, Oxford OX3 0BP, UK; sophie.manson_2019@brookes.ac.uk (S.M.); anekaris@brookes.ac.uk (K.A.I.N.); mbalestri@brookes.ac.uk (M.B.); vnijman@brookes.ac.uk (V.N.)
2  Little Fireface Project, Cipaganti, Bandung 40131, Indonesia; katey.hedger@gmail.com (K.H.); ahmadnabilff@gmail.com (N.A.); estheradinda810@gmail.com (E.A.)
3  Faculty of Forestry, Universitas Gadjah Mada, Yogyakarta 55281, Indonesia; budiadi@ugm.ac.id (B.B.); maimron@ugm.ac.id (M.A.I.)
*  Correspondence: mcampera@brookes.ac.uk

**Abstract:** Pollination services, from both wild and managed populations of insect pollinators, have degraded as a result of agricultural intensification. Whilst 75% of economically important crops depend on insect pollinators for cultivation, 40% of insect pollinator species are threatened with extinction. Pollination services must be preserved if there is to be enough food for a global population whose demand is expected to double, if not triple, by 2050. Pollinator diversity and pollinator efficiency have been found to increase as a result of wildlife-friendly farming practices (i.e., natural chemicals and fertilizers and agroforestry). We evaluated the presence of insect pollinators in 42 coffee home gardens in West Java, Indonesia. Via generalized linear mixed models, we found that number of visitor species (β = 0.418 ± SE 0.194, $p$ = 0.031) and visitation time (β = 0.845 ± SE 0.308, $p$ = 0.006) decreased as farms were more intensely managed, (i.e., used chemical pesticides), compared to fields using organic practices. As knowledge of pollination services is widespread amongst smallholder farmers in Indonesia and beyond due to the long-held tradition of beekeeping, these results will add to their existing knowledge and empower farmers to enhance resources for pollinator species through agroforestry and natural pest management. Although we found significant differences in pollination services provided in intensely managed and wildlife-friendly farms, chemical use can affect farms far beyond a particular area of production. Therefore, pollinator conservation must be applied at a landscape level and involve all stakeholders, including farmers, when making effective policies.

**Keywords:** wildlife-friendly; pollination; climate change; butterflies; bees; syrphid flies; Indonesia; ecosystem services; organic; agricultural intensification

## 1. Introduction

As of now, 75% of economically important food crops depend on insect pollinators for reproduction purposes [1]. It is this dependence on insect pollinators that forms the intrinsic link between ecosystem services and food security. As the human population is set to increase to 10 billion by 2050, within this timeline, global food demand is expected to at least double [1–4]. In order to ensure that enough food is produced for a growing population, and produced equitably, over the last century there has been a global shift from traditional, organic farming practices to intensive agriculture that maximizes production [5]. This intensification comprised the use of chemical pesticides, herbicides and fertilizers, the creation of monocultures, and the management of pollination services [6]. Both wild and managed pollinators have suffered throughout this transition to intensive agriculture, with 40% of invertebrate pollinators at risk of extinction [7].

Pollinator species around the globe have become increasingly threatened by intensive farming practices and climate change, with both directly affecting pollinator physiology, phenology, and behavior, whilst also altering the ecosystems in which pollinators reside [8–10]. Intensive farming threatens pollinators in two ways: (1) the prophylactic use of broad-spectrum chemical pesticides; and (2) the destruction of pollinator habitat for monocultures [11–14]. Chemical pesticides are commonly broad-spectrum, meaning that they are toxic to a broad range of insect species, including pollinator species [15]. As a result, it is common that non-target insect populations are also affected in myriad ways, spanning direct physiological and behavioral implications and indirect stressors caused by co-exposure [12,16–18]. In addition, although it is chemical pesticides that are primarily harmful to insects, it is true that some biopesticides (i.e., naturally-derived pesticides), can be broad-spectrum and, as a result, also damaging to non-target insect species [19–22]. Not only do non-target insect species often provide pollination services, but they provide other beneficial services, such as pest control by natural enemies [23,24]. The intensive nature of monocultures, particularly for mass-flowering crops, can potentially encourage pollinator visitation time, with pollinator efficiency sometimes being higher in monocultures than it is in polycultures [25,26]. However, when land is cleared for monocultures, nesting, larval, and floral resources for insect pollinator species are diminished [14]. Due to the lack of resources found in monocultures, and the often insufficient resources surrounding them, it is common for studies to observe reduced pollinator abundance and, in turn, fruit set in monocultures [13,14]. This is in direct competition with the commonly perpetuated idea that intensive farming methods are adopted to maximize productivity, leading to widespread reluctance to adopt organic farming practices [27].

In addition to monocultures and chemical pesticides, climate change poses huge risks to pollinator phenology [28,29]. West Java, Indonesia has been identified as one of the areas in Southeast Asia that is most vulnerable to climate change, causing rains to become more unpredictable and, in recent years, leading to extended dry periods and droughts [30–32]. Mean temperature in this area has also increased, something which has been further exacerbated by deforestation [33]. In Java, Indonesia, the most commonly managed species of bee is *Apis mellifera*, the western honey bee, as the island's flowering calendar previously fell in line with that of *A. mellifera* [34,35]. However, *A. mellifera* has not been able to adapt to changing climatic conditions, as temperature increases brought on by climate change have caused phenological changes in plant-pollinator interactions, causing mismatches in blooming and insect emergence [28,29,34,35]. Whilst *A. mellifera* populations are being threatened by climate change, *A. mellifera* can also stifle the diversity of wild pollinators present in a particular area [36]. *A. mellifera* live in large colonies, giving them a competitive advantage over other species of stingless and solitary bees due to their ability to recruit colony members to foraging resources [36]. Considering the increasing difficulty to manage *A. mellifera* for pollination purposes, and the potential for *A. mellifera* to further threaten wild bee populations, there is an urgent requirement to preserve pollinator diversity, as phenological synchrony is more likely to be achieved through species complementarity [37,38]. Species complementarity is defined as the presence of several species residing in separate functional niches combining to provide a superior overall service, such as pollination [38,39].

Pollinator conservation will depend largely on landscape arrangement, farming intensity and climate. However, organic, wildlife-friendly farming practices (i.e., no chemicals, low intensity, shade cover, and tree diversity) have been proven to preserve pollinator abundance, diversity, visitation time, and fruit set across many environments [2,14,39–41]. In West Java, Indonesia, due to historic success with implementing wildlife-friendly farming methods, such as the use of biopesticides (i.e., naturally derived pesticides), there is already a culture of using traditional farming practices in smallholder farms. Integrated pest management (IPM), a global initiative that began in the 1970s to address pesticide resistance and the deleterious effects of chemical pesticides through the integration of holistic, naturally-derived farming practices, was particularly successful in Indonesia [5,42,43]. One

possible reason for its success in Indonesia may be the presence of farming cooperatives, groups of farmers in close communities that aid the dissemination of information, tools and funding [44]. In some areas of Indonesia, chemical pesticide usage declined by 96% [5]. However, in 1999, IPM in Indonesia formally ended, and therefore companies were able to start selling and subsidizing chemical pesticides once again.

Indonesia is the fourth largest producer of coffee (*Coffea* spp.) in the world, with 96% of this coffee coming from smallholder farms [45]. At our study site in West Java, Indonesia, Arabica coffee (*Coffea arabica*) is the most commonly cultivated crop. Coffee farms exist within an agroforest matrix, meaning that the natural forest, to a varying extent, is present within and around these farms [44]. Around the world, coffee is often shade-grown to protect crops from the sun. However, shade cover and shade tree diversity have been found to positively influence the provision of ecosystem services, such as pest control by natural enemies and pollination [39]. Although chemical pesticide usage has increased due to their price and ease of accessibility in comparison to biopesticides following the end of IPM, traditional farming practices, such as non-chemical fertilizers/pesticides, have persisted. As a result, within our study site, there is variation in chemical use, producing a varied landscape with regard to farm management intensity [44].

Prioritizing the conservation of pollinator species and preserving pollination provision is particularly critical if we are to ensure food for all in a planet with a growing population [1–4]. In West Java, coffee is grown alongside other economically important crops, such as cabbage (*Brassica oleracea*) and chayote (*Sechium edule*) [44]. Therefore, organic practices that are encouraged for coffee certifications, such as Wildlife Friendly™ and Organic™, are also positively influencing the provision of ecosystem services for the other crops in these farms. In addition, although coffee cannot be used as a food source, demand for coffee is growing and demand is unlikely to wane in coming years. Additionally, considering it is one of the most economically important crops for Indonesian farmers, coffee farming may not be providing sustenance but it is providing an income for 25 million farmers across the world [44–46].

In this study, we assessed pollinator diversity and pollinator visitation time in coffee farms of varying management intensities in an agroforest ecosystem in West Java, Indonesia. Our aim was to investigate the effect of management intensity (i.e., chemical usage, shade cover and shade tree diversity, and temperature) on the provision and reception of pollinator services. Due to the widely documented impacts of chemical usage on pollinator physiology and behavior discussed previously, we expected pollinator diversity and pollinator visitation time to decrease with increased chemical usage. Conversely, we expected pollinator diversity and visitation time to increase with increased shade cover and shade tree diversity. Finally, we expected pollination visitation time to increase with increasing temperature, in line with Arroyo et al.'s [47] finding that pollinator activity reduced at lower temperatures. However, we do expect that pollinator visitation time would decrease with extreme temperatures. With our results, we hope to empower farmers with the information necessary to make environmentally conscious decisions with regards to the preservation of pollination services. Additionally, we hope to better understand the effect of climate change on pollination services, thus helping farmers to adapt to increasing temperatures and the potential ramifications for their crops.

## 2. Materials and Methods

### 2.1. Study Site

We collected data from 42 coffee home gardens in the municipalities of Cipaganti and Pangauban, Garut Regency, West Java, Indonesia (7.2786° S, 107.7577° E). These home gardens, further referred to as farms, are smallholder owned and exist within an agroforest matrix (i.e., the natural habitat is maintained within and around farms and crops are rotated annually) [44,48]. Farmers plant coffee in tandem with other economically important crops, such as cabbage, chilli (*Capsicum frutescens*), cassava (*Manihot esculenta*) or underneath chayote [44]. Farms are situated between 1105 and 2105 m above sea level, with

a maximum distance of 1805 m and a minimum distance of 15 m away from each other [49]. West Java does not experience strict seasons. However, heavier rainfall is observed between December and April, although this is changing year on year due to climate change [49]. Coffee plants bloom twice a year (i.e., April and November), with the bloom in November producing a smaller fruit set.

*2.2. Data Collection*

We collected data in coffee fields when the majority of coffee plants were blooming (i.e., >50% of the total plants in the field are blooming, and when each plant that has flowers is covered by more than 50% by blooming flowers). Since the blooming period is very narrow for coffee plants, we managed to collect data once on 42 fields over 4 blooming periods (July–August 2019, 2020; November–December 2020, 2021). During the blooming period, and when not raining, a team composed of one researcher (N.A. or E.A.) surveyed all of the fields between 8:00 and 13:00 h in search for fields where the majority of coffee plants were blooming. It took four blooming periods to complete all of the 42 fields. When we encountered a suitable field, the researcher started the data collection by selecting five random plants in the field. The researcher than started collecting data for 10 min in each plant, totaling 50 min of observation in each field [50]. Before the data collection period, we identified and catalogued the species of butterflies, bees, wasps, and syrphid flies present in the area based on inventories regularly carried out by the Little Fireface Project between 2012 and 2019. We created a list of species with images to allow quick identification. For new species missing from the list, we described a morphospecies and took a picture for further identification. During the data collection time, we also used HOBO temperature logger and the app HOBOmobile to record temperature and humidity every minute during the 10-min sampling period. We than calculated an average temperature and humidity for each field.

In each field, we estimated the shade cover via Canopeo App that calculates the proportion of area shaded from pictures [49,51] (Figure 1). We also collected information on the richness (i.e., total number of species) of shade trees. We estimated the use of agrochemicals (i.e., fertilizers urea and NPK and pesticide Endosulfan) by farmers via interviews. We categorized the use of agrochemicals as: (1) no chemicals used; (2) chemical fertilizers and pesticides mixed with organic products; and (3) intensive use of chemical fertilizers and pesticides, no organic materials used. More details on the data collection can be found in Campera et al. [49].

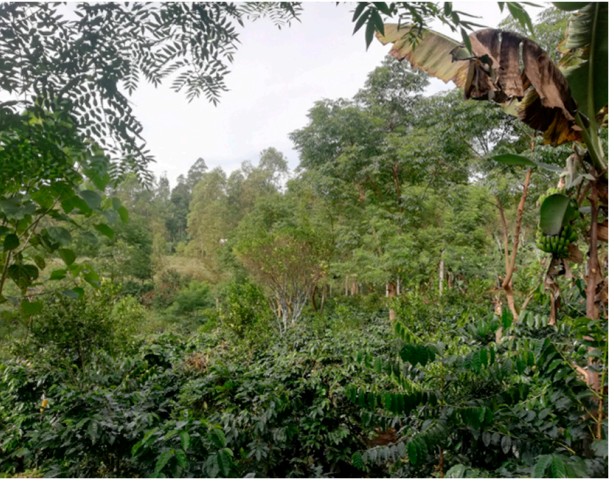
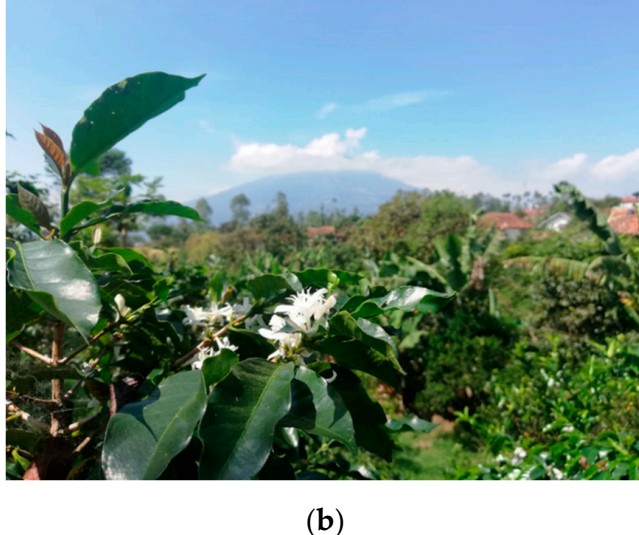

(**a**)  (**b**)

**Figure 1.** Examples of coffee home gardens that can be found at the study site. (**a**) coffee home garden with partial shade given by multiple tree species; (**b**) coffee home garden with no shade.

*2.3. Data Analysis*

We tested whether number of visitor species and pollinator visitation time (i.e., proportion of time when the flowers were visited by pollinators/total observation time) were influenced by use of chemicals, humidity, temperature, shade cover, and shade tree richness via generalized linear mixed models. We used the "glmmTMB" function in the "glmmTMB" package as this function includes several fit families that are suitable to deal with counts and proportions [52]. We tested different fit functions (the ones that are included in the package) and included or excluded a zero-inflation term based on the QQ plot residuals and residual vs predicted plot from the package "DHARMa" [53]. The selected fit functions were genpois for the number of visitor species and zero inflated beta for pollinator visitation time. We included the data collection period (i.e., the four blooming periods) as a random effect in the model. We ran pairwise contrasts using a Bonferroni-Holm post hoc correction via the function "emmeans" in the package "emmeans". We considered $p = 0.05$ as level of significance. We ran all of the analyses with R v 4.1.0.

## 3. Results

A total of 29 visitor species were observed to pollinator coffee plants, totaling 5016 s of pollination (~4% of the total observation time). Lepidoptera were the most frequent visitor with a total of 2264 s of pollinator visitation time, followed by Hymenoptera (1547 s) and Diptera (1205 s) (Table 1). The number of visitor species was higher in organic fields (mean: 4.04 species per field; 95% CI: 3.34–4.89 species per field) than in fields with intensive chemical use (mean: 2.66 species per field; 95% CI: 2.17–3.27 species per field) (Figure 2; Table 2). Pollinator visitation time was also significantly higher in organic fields (mean: 3.7% of time; 95% CI: 2.6–5.2% of time) than in fields with intensive chemical use (mean: 1.6% of time; 95% CI: 1.1–2.4% of time) (Figure 3; Table 2).

Pollinator visitation time was also influenced by the mean temperature during the data collection, with a positive influence of increased temperature on the pollinator visitation time in coffee plants (Figure 4; Table 2). The other predictors did not influence the response variables, only influencing a trend towards a significance positive relationship between humidity and pollinator visitation time (Table 2).

**Table 1.** Number of pollinator species identified in 42 coffee fields in Cipaganti and Pangauban, West Java. Total observation time was 35 h during four blooming seasons (July–August 2019, 2020; November–December 2020, 2021).

| Order | Family | N Species | Pollinator Visitation Time (s) |
|---|---|---|---|
| Lepidoptera | Nymphalidae | 5 | 1636 |
| | Papilionidae | 5 | 560 |
| | Pieridae | 1 | 65 |
| | Lycaenidae | 1 | 3 |
| Hymenoptera | Apidae | 8 | 1336 |
| | Vespidae | 2 | 211 |
| Diptera | Syrphidae | 5 | 1194 |
| | Drosophilidae | 2 | 11 |
| Total | | 29 | 5016 |

**Table 2.** Results of the generalized linear mixed models with number of visitor species and pollinator visitation time in coffee plants as response variables. Data are collected on 42 fields in the municipalities of Cipaganti and Pangauban (West Java) during four blooming seasons (July–August 2019, 2020; November–December 2020, 2021).

| Response | Predictor | Estimate | St. Error | Z-Value | *p*-Value |
|---|---|---|---|---|---|
| Number of visitor species [a] | Intercept | 0.366 | 1.225 | 0.299 | 0.765 |
| | Chemicals: mixed [c] | 0.129 | 0.192 | 0.672 | 0.501 |
| | Chemicals: organic [c] | 0.418 | 0.194 | 2.152 * | 0.031 |
| | Humidity (%) | 0.004 | 0.006 | 0.567 | 0.571 |
| | Shade cover (%) | 0.001 | 0.005 | 0.112 | 0.911 |
| | Shade tree richness | 0.029 | 0.046 | 0.619 | 0.536 |
| | Temperature (°C) | 0.011 | 0.039 | 0.293 | 0.769 |
| Pollinator visitation time [b] | Intercept | −8.905 | 1.754 | −5.077 * | <0.001 |
| | Chemicals: mixed [c] | 0.580 | 0.266 | 2.183 * | 0.029 |
| | Chemicals: organic [c] | 0.845 | 0.308 | 2.745 * | 0.006 |
| | Humidity (%) | 0.015 | 0.008 | 1.900 | 0.057 |
| | Shade cover (%) | 0.003 | 0.007 | 0.486 | 0.627 |
| | Shade tree richness | −0.002 | 0.070 | −0.034 | 0.973 |
| | Temperature (°C) | 0.152 | 0.056 | 2.709 * | 0.007 |

[a] family fit, genpois; [b] family fit, zero inflated beta; [c] reference category, intensive; * $p < 0.05$.

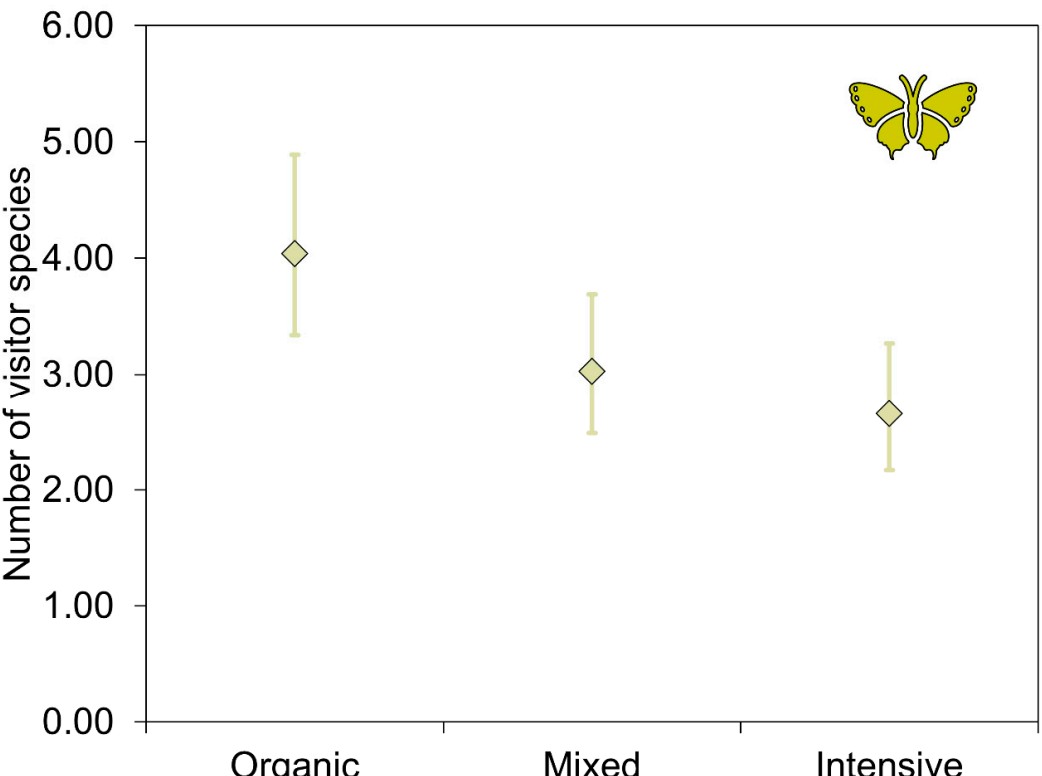

**Figure 2.** Estimated marginal means and 95% CI for the predictor chemical use based on a generalized linear mixed model with the number of visitor species as dependent variable. Data are collected on 42 fields in the municipalities of Cipaganti and Pangauban (West Java) during four blooming seasons (July–August 2019, 2020; November–December 2020, 2021).

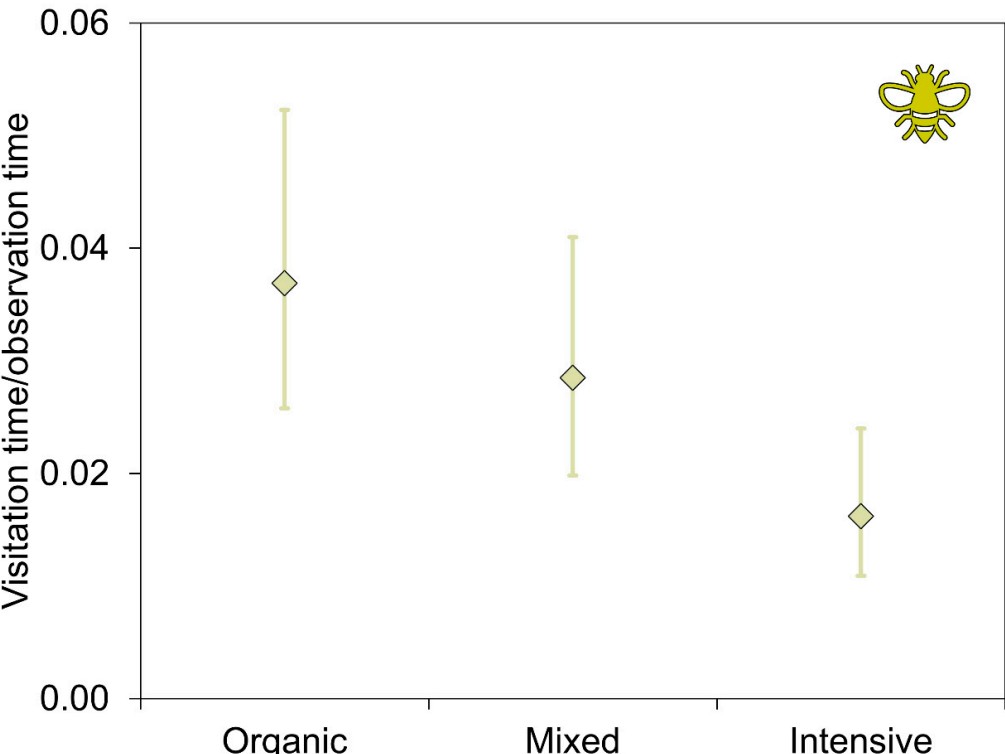

**Figure 3.** Estimated marginal means and 95% CI for the predictor chemical use based on a generalized linear mixed model with the ratio pollinator visitation time over total observation time as dependent variable. Data are collected on 42 fields in the municipalities of Cipaganti and Pangauban (West Java) during four blooming seasons (July–August 2019, 2020; November–December 2020, 2021).

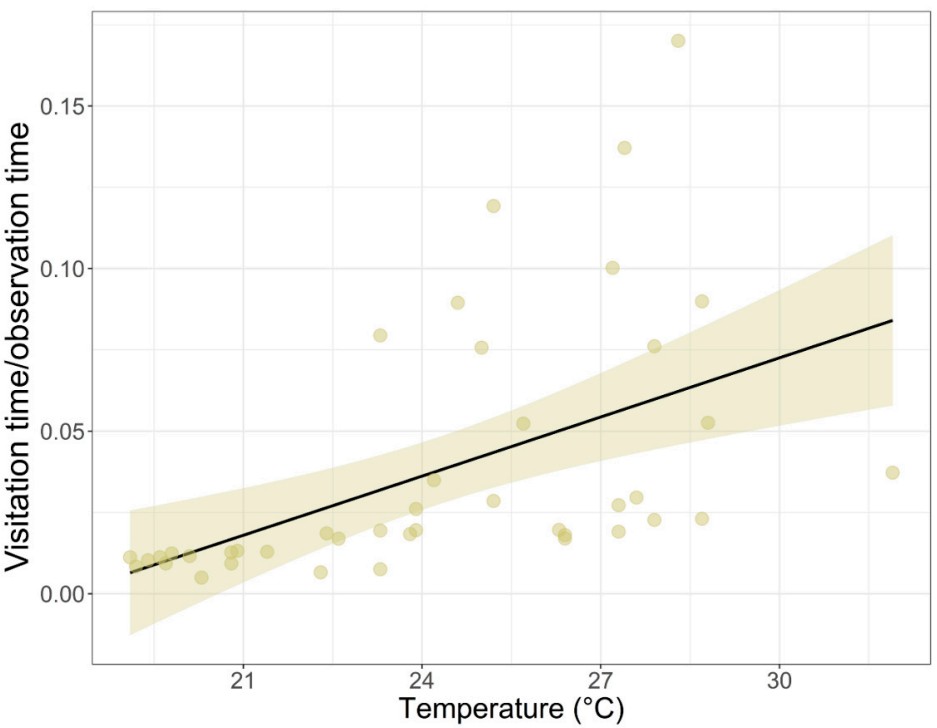

**Figure 4.** Significant relationship between temperature and pollinator visitation time in coffee plants. Data are collected on 42 fields in the municipalities of Cipaganti and Pangauban (West Java) during four blooming seasons (July–August 2019, 2020; November–December 2020, 2021). Shaded areas represent 95% CI.

## 4. Discussion

We found that pollinator visitation time was significantly affected by farming intensity, with decreased pollinator visitation times in farms that used a mix of chemicals and organic practices and further decreased pollinator visitation times in farms that used exclusively chemicals. Increased pollinator visitation time has been found to be significant when measuring the efficiency of pollination attempts (i.e., pollination receipt) [54]. Although we define "farming intensity" in terms of chemical usage, studies often contribute to the definition of this term by measuring the complexity of farms and surrounding land (i.e., landscape heterogeneity) [8,55]. It is widely documented that farms that are more intensely managed, with regards to landscape heterogeneity, experience reduced pollination services [8,41,55,56]. Although, Andersson et al. [55] state that for studies that use both chemical usage and landscape heterogeneity to define farming intensity, it is difficult to disentangle whether chemical usage or landscape heterogeneity affect pollinator visitation more. In our study, shade cover and shade tree diversity did not affect pollinator visitation time, yet we found that chemical usage did. Therefore, our study is a useful contribution to the body of literature documenting the effects of varying farming practices on pollinator visitation and will help to disentangle what practices are most damaging to certain ecosystem services.

Furthermore, we found that there was lower pollinator diversity in farms that were more intensely managed (i.e., used exclusively chemicals or used some chemicals to farm). Our finding is in line with existing studies that have found that intensive farming practices in general, not only chemical usage, contribute to lower pollinator diversity, particularly in tropical environments [11,56–59]. High pollinator diversity has been shown to improve crop productivity in smallholder farms, increase coffee fruit set and improve the amount and frequency of pollination services [50,60–62]. Consequently, preserving pollinator diversity is essential if we are to address global food insecurities, reverse global pollinator decline and re-establish degraded ecosystem service provision [2,63].

In addition to farming intensity, pollination services were also found to be significantly affected by temperature, with pollinator visitation time increasing with increased temperature. In theory, and as we can see from our results, temperature increase often positively influences pollination services in the short-term by increasing pollinator visitation time [64]. However, the optimal temperature for pollination changes dependent on other environmental variables, particularly altitude [47,65]. Furthermore, we found a linear relationship between temperature and pollinator visitation time up to 31 °C, with only one point above 29 °C. But with regards to extreme temperature above 30 °C, we would expect a decrease in pollinator visitation time. In addition to the threat of extreme temperatures, it is the effect of climate change on pollinator phenology that will also significantly disturb plant-pollinator interactions [28,29]. Increased pollinator diversity, and ensuring species complementarity, will reduce the likelihood of phenological asynchronies and will bolster an ecosystem against the deleterious effects of climate change on pollination [37,38]. Furthermore, in future studies, temperature should be measured in relation to altitude to establish optimal pollination conditions for farms at varying elevations due to the increased importance of Lepidopterans and Dipterans at higher elevations [65]. Smallholder farmers in tropical countries are forecast to be one of the worst hit demographics in terms of climate change [32]. With this in mind, it is important that the long-term effects of temperature increase are monitored, in line with pollinator species emergence, to be able to inform smallholder farmers of potential changes to ecosystem services.

Shade cover and shade tree richness did not significantly affect pollinator visitation time or pollinator diversity within farms. On one hand, shade cover within agroforest ecosystems has been praised within the literature as providing an environment that sustains and encourages biodiversity, with regards to both pollinator and non-pollinator species [49,66,67]. However, there has been discussion as to whether shade tree richness can negatively impact pollination services due to crop plants having to compete with flowering shade trees for pollinator visitation [25,68,69]. In addition, increased shade tree

richness can encourage pollinators to diversify the species they visit, meaning that pollen deposits in agroforest ecosystems can be less than what is observed in [25]. Prado et al. [69] found that for coffee in particular, shade trees did not detract from pollinator visitation and furthermore, in line with our results, found that the presence of shade trees was insignificant when determining pollinator visitation. Additionally, social bee pollinator species of the family Apidae (Hymenoptera: Apidae), comprising honey bees, bumblebees, and carpenter bees (*Xylocopa* sp.) made up the majority of species visiting coffee farms. Hennig [68] found that when crop plants relied on social bee species for the majority of their pollination services their pollination was less affected by co-occurring plant species.

It is clear from our results that farming intensity is a major determinant of the provision of pollination services, the duration of pollination and the diversity of pollinators present, all of which are negatively influenced by the use of chemicals. Wildlife-friendly farming methods have the potential to reduce negative impacts of farming on pollinator populations and pollination provision, both directly and indirectly, with biodiversity itself proven to enhance plant-pollinator interactions [37,50,55,56,70]. This is shown aptly by Klein [71] who found that pollinator richness was higher in agroforest environments than within closed forest, showing that wildlife-friendly farming practices are capable of hosting higher than expected pollinator diversity. Whilst Hardman et al. [70] found that some ecosystem services were compromised when pollination provision is prioritized (e.g., lower crop yields in wildlife-friendly, organic systems), Pywell et al. [72] found that crop yields increased following the creation of habitat to enhance pollination services. Differences in how wildlife-friendly farming affects crop yields depends largely on the crop being harvested. In our case, and in the case of the Pywell et al. [72] study, farmers use a system of land-sharing (i.e., agroforestry), as opposed to land-sparing which intensifies farming in one area whilst sparing other areas for entirely conservation purposes [73]. It could be extrapolated that for areas in which land-sharing is not an option, crop yields may suffer as a result of implementing wildlife-friendly practices to preserve pollination services, such as what is observed in Hardman et al.'s [62] study. However, whilst crop yields may decline as a result of implementing wildlife-friendly farming practices in few cases, it could be argued that this effect is diminished due to increased pollinator diversity and complementarity providing greater ecosystem resilience and thus, greater security with regard to food production and farmer income [37,38].

Although our study reinforces the use of wildlife-friendly farming practices and their positive effects on pollinator activity, other studies have found that wildlife-friendly farming practices are not without their disadvantages. Biopesticides are naturally-derived alternatives to broad-spectrum chemical pesticides and use either plants, fungi or hormones to target pest species [19,20]. Amongst their many benefits, perhaps the most significant justification for their use is the fact that the user can target a particular pest species rather than endangering insects in general, including pollinators. As biopesticides increase in popularity, there will inevitably be opportunities to cut corners. Some biopesticides have been used for broad-spectrum applications, an example of which is Spinosad, a pesticide derived from a soil bacterium toxic to insects. Although it is naturally derived, its toxicity is widespread and it has been found to negatively affect the behavior and physiology of pollinators [21,22]. Therefore, whilst the use of wildlife-friendly farming practices, such as biopesticides, is essential for the preservation of biodiversity, pollinator diversity particularly, farming practices should be proven to be species specific with the safety of pollinators prioritized.

The results of this study will help to communicate the importance of wildlife, and the ecosystem services they provide, to farmers, empowering them to continue using wildlife-friendly farming practices. Globally, there is a general reluctance to adopt organic methods due to expense, misinformation regarding productivity and extra labor [27]. Monocultures have been shown and publicized to be more productive, in all environments, with land clearing and chemical input being used in combination to increase short-term profits [74,75]. We have shown the importance of wildlife-friendly practices in attracting pollinators, but

it is important that we can also prove the importance of pollinators for coffee fruit set, inevitably one of the priorities of smallholder farmers. Moving forward, we will continue to assess the presence and diversity of pollinators within coffee farms and how this, alongside the use of wildlife-friendly farming practices, influences the productivity of coffee plants and income of farmers.

Finally, although we observed large variation in pollinator diversity and visitation time between farms that used chemicals and those that practiced organic methods, pollinator conservation has to be applied at a landscape-scale if it is to protect pollinator species in the long-term [40,58]. Pollinators can be affected by chemical pesticides used in areas that far exceed a particular area of production, both spatially and temporally. Krupke et al. [17] explained these interactions in both temporal and spatial terms. With regards to distance, pesticides can prevail in technically untreated areas due to the movement of treated soil and seed and soil dust following largescale planting. Furthermore, this can be soil and dust that has not been recently treated, due to the ability of chemical pesticides to prevail within an environment long after its application [17]. Knowledge of the preservation of pollinator nesting sites and floral resources is relatively well known globally, particularly within smallholder farming communities that had previously relied on traditional farming practices, such as beekeeping [40]. However, the adoption of wildlife-friendly practices is in direct conflict with the current trend of simplification in farming, with accessibility, price and time all contributing to the farming practices that farmers must use [40]. Therefore, whilst we must translate existing knowledge of smallholder farming communities to effective policies, the culture and behavior surrounding farming, and the rhetoric around intensification for optimum production, must change in order for new policies to be successful.

## 5. Conclusions

In our study, we found that pollinator diversity and pollinator visitation time declined with increasing chemical usage in coffee farms of varying management intensity in West Java, Indonesia. Farms that used wildlife-friendly methods, such as natural pesticides and fertilizers, observed significantly higher pollinator diversity and visitation time, indicating that wildlife-friendly methods preserve ecosystem services. By preserving the provision of ecosystem services, such as pollination, farmers are able to ensure the long-term sustainability of their crops, thus removing their reliance on managed populations of bees. In addition, we found that pollinator visitation time increased with increasing temperature, helping to further demonstrate how climate change is affecting the behaviors of pollinators. With increasing temperature, plant-pollinator interactions are expected to degrade as a result of changes in pollinator and plant phenology, rendering the use of managed, non-native bee populations less feasible. The dissemination of our results to smallholder coffee cooperatives in our study area will build on existing knowledge of plant-pollinator interactions, further empowering farmers to invest in wildlife-friendly farming practices. We hope to encourage collaboration between farming cooperatives and local government organizations to promote the benefits of wildlife-friendly farming practices for biodiversity, for farmers and for the security of generations to come.

**Author Contributions:** Conceptualization, K.A.I.N. and M.C.; methodology, K.H. and M.C.; software, M.C.; validation, K.H. and M.C.; formal analysis, M.C.; investigation, N.A. and E.A.; resources, K.A.I.N.; data curation, S.M., K.A.I.N., K.H., M.B. and M.C.; writing—original draft preparation, S.M. and M.C.; writing—review and editing, K.H., K.A.I.N., M.B., B.B., M.A.I. and V.N.; visualization, M.C.; supervision, K.A.I.N., K.H., M.A.I., B.B., V.N. and M.C.; project administration, K.A.I.N. and K.H.; funding acquisition, K.A.I.N. and M.B. All authors have read and agreed to the published version of the manuscript.

**Funding:** This research was funded by Augsburg Zoo, Brevard Zoo, Cleveland Zoo and Zoo Society, Columbus Zoo and Aquarium, Disney Worldwide Conservation Fund, Global Challenges Fund, Henry Doorly Zoo, International Primate Protection League, Little Fireface Project, Mohamed bin Zayed Species Conservation Fund (152511813, 182519928), Margot Marsh Biodiversity Fund, Memphis Zoo, Moody Gardens Zoo, Paradise Wildlife Park, People's Trust for Endangered Species, Phoenix Zoo, Primate Action Fund, Shaldon Wildlife Trust, Sophie Danforth Conservation Biology Fund, and ZGAP.

**Institutional Review Board Statement:** The study was conducted in accordance with the Declaration of Helsinki. The interviews with farmers were approved by the Oxford Brookes University Ethics Committee (number 181256).

**Informed Consent Statement:** Informed consent was obtained from all subjects involved in the study.

**Data Availability Statement:** The data presented in this study are available on request from the corresponding author.

**Acknowledgments:** We thank Indonesia RISTEK for authorizing the study (research permit 24/FRP/E.5/Dit.KI/I/2019).

**Conflicts of Interest:** The authors declare no conflict of interest. The funders had no role in the design of the study; in the collection, analyses, or interpretation of data; in the writing of the manuscript; or in the decision to publish the results.

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
