# Peer review of "Flower Visitation Time and Number of Visitor Species Are Reduced by the Use of Agrochemicals in Coffee Home Gardens"

_agronomy, doi:10.3390/agronomy12020509_

Round 1

Reviewer 1 Report

This is an interesting study with some good merit. It would be suitable for publication given the following changes. All comments should be carefully addressed but I have stressed those that are of critical importance and MUST be addressed before publication is considered. 

  1. The abstract is too long and should be reduced by 30-40%. The first line can be removed. The results section can be shortened to the key results only.
  2. Much of the first paragraph of the introduction could also be removed. I suggest starting at line 48 instead. 
  3. line 64 - "meaning that they can be used to manage several insect pest species". I think, "they are toxic to a broad range of insect species" would be more accurate. \
  4. line 66 "caught in the crossfire" is too informal. what do you mean specifically? 
  5. line 68 - "Theoretically, monocultures are not inherently harmful to pollinators" statement needs supporting citation. Furthermore, I think you mean that some pollinators benefit from monocultures, which is not what you are saying. Be more precise. 
  6. line 79-86. The meaning here is not clear or obviously relevant. I suggest removing or changing substantially. 
  7. More information on farming practices for coffee production would greatly benefit the introduction, with particular reference to IPM. 
  8. line 122-125- " We expect that pollinator diversity and pollinator visitation time will decrease with increased chemical usage. Conversely, we expect that pollinator diversity and visitation time will increase with increased shade cover and shade tree diversity. Finally, we expect pollination visitation time to increase with increasing temperature" - You have not explained your reasoning for these hypotheses. You MUST address this in your intro. 
  9. line 126 - "we hope to show" suggests you are not scientifically impartial but are looking for a particular result. Please reword. 
  10. line 170 - You say you have given more details in a separate publication but given that these details are critical to the study they MUST be included here. Provide more details about the kinds of chemicals used, the reasons for their use, as well as the the threshold of applications between intensive and mixed. 
  11. Some pictures of the gardens/ farms and flower visitors would be great. 
  12. figure 1. I'm not familiar with estimated marginal means, I presume the error bars show confidence intervals. if so, amend the figure to communicate this more clearly. 
  13. figure 3. I don't think fitting a straight line is appropriate here. The line suggests that increasing temperature will increase flower visitation. However, your data and common sense would suggest that at higher temperatures (>35) visitation will decrease. As such, a linear relationship is inappropriate here. You MUST address this. 
  14. I do not like the use of the term "pollination time". Strictly speaking, pollination time infers the transfer of pollen. You did not measure this. You measured flower visitation time/ duration which is different because you do not know if the visits resulted in pollination. You MUST change throughout. 
  15. line 244 smallholder farms not farmers
  16. line 311 to 319 - I think you need to make this distinction at the start of the manuscript; even "natural pesticides" can have broad spectrum activity and can be harmful to beneficial insects. 
  17. Conclusions are too long. Reduce by at least 50%. Focus on how your findings relate to your original aims and questions. 
  18. Reducing pesticide use could also help with other aspects of food production including pest control by natural enemies. I do not believe you have mentioned this. 

Author Response

This is an interesting study with some good merit. It would be suitable for publication given the following changes. All comments should be carefully addressed but I have stressed those that are of critical importance and MUST be addressed before publication is considered. 

  1. The abstract is too long and should be reduced by 30-40%. The first line can be removed. The results section can be shortened to the key results only.

We appreciate this comment as it is often the reviewer who can spot what is essential information for an abstract rather than the author. We have removed the first two sentences and have removed the result surrounding temperature and pollinator visitation time and its subsequent discussion.

  1. Much of the first paragraph of the introduction could also be removed. I suggest starting at line 48 instead.

We agree that the beginning two sentences are not necessary to the rest of the paper, these have been omitted.

  1. line 64 - "meaning that they can be used to manage several insect pest species". I think, "they are toxic to a broad range of insect species" would be more accurate.

We agree that your suggestion lays a better context and acts as a better definition. Your suggestion has replaced the original.

  1. line 66 "caught in the crossfire" is too informal. what do you mean specifically?

We agree that this is too informal for this paper and, as a result, have changed it to the following: “As a result, it is common that non-target insect populations are also affected in myriad ways, spanning direct physiological and behavioral implications and indirect stressors caused by co-exposure.”

  1. line 68 - "Theoretically, monocultures are not inherently harmful to pollinators" statement needs supporting citation. Furthermore, I think you mean that some pollinators benefit from monocultures, which is not what you are saying. Be more precise.

We agree that this could be a potentially misleading statement, so we have changed the above sentence to the following: “The intensive nature of monocultures, particularly for mass-flowering crops, can potentially encourage pollinator visitation time, with pollinator efficiency sometimes being higher in monocultures than it is in polycultures.”

  1. line 79-86. The meaning here is not clear or obviously relevant. I suggest removing or changing substantially.

We agree that this is unnecessary context and have removed the entirety of the suggested paragraph.

  1. More information on farming practices for coffee production would greatly benefit the introduction, with particular reference to IPM.

We agree that not enough context with regard to coffee agriculture was given, therefore, we have added a paragraph in the introduction to define coffee agriculture in relation to Indonesia and IPM. This paragraph begins at line 138 and finishes at line 149.

  1. line 122-125- " We expect that pollinator diversity and pollinator visitation time will decrease with increased chemical usage. Conversely, we expect that pollinator diversity and visitation time will increase with increased shade cover and shade tree diversity. Finally, we expect pollination visitation time to increase with increasing temperature" - You have not explained your reasoning for these hypotheses. You MUST address this in your intro.

We are in agreement that these hypotheses were not fully justified within the intro, particularly regarding shade cover and shade tree diversity.

    • For the first hypothesis, we added the following to the beginning of the sentence: “Due to the widely documented impacts of chemical usage on pollinator physiology and behavior discussed previously…”.
    • For the second hypothesis, we have elaborated on shade cover and shade tree diversity several times in the introduction, with a detailed explanation from line 142-145.
    • For the third hypothesis, we have provided more context with regard to the effect of climate change on pollinator phenology and behavior from line 93-116. This involved moving part of the discussion to the introduction.

  1. line 126 - "we hope to show" suggests you are not scientifically impartial but are looking for a particular result. Please reword.

We agree that this indicates that we are looking for a particular result. We have changed the above sentence to the following: “With our results, we hope to empower farmers with the information necessary to make environmentally conscious decisions with regards to the preservation of pollination services.”

  1. line 170 - You say you have given more details in a separate publication but given that these details are critical to the study they MUST be included here. Provide more details about the kinds of chemicals used, the reasons for their use, as well as the the threshold of applications between intensive and mixed.

We have now added the information about chemicals used. We did not use a threshold as that would depend on the area of the garden. We included the category mixed when both organic and chemical fertilisers and pesticides were used.

  1. Some pictures of the gardens/ farms and flower visitors would be great.

We agree that this would be a useful addition and have added photos accordingly (figure 1). 

  1. figure 1. I'm not familiar with estimated marginal means, I presume the error bars show confidence intervals. if so, amend the figure to communicate this more clearly. 

We have now clarified that error bars are 95% CI. Thanks for spotting this

  1. figure 3. I don't think fitting a straight line is appropriate here. The line suggests that increasing temperature will increase flower visitation. However, your data and common sense would suggest that at higher temperatures (>35) visitation will decrease. As such, a linear relationship is inappropriate here. You MUST address this.

We have taken this comment into consideration and have decided to make it clearer in the discussion that our results only took into account temperatures up to 31ËšC. Therefore, a linear relationship is appropriate for this dataset, but we’d expect pollinator visitation time to decline in extreme temperatures (>30 ËšC).

  1. I do not like the use of the term "pollination time". Strictly speaking, pollination time infers the transfer of pollen. You did not measure this. You measured flower visitation time/ duration which is different because you do not know if the visits resulted in pollination. You MUST change throughout.

We agree that this is potentially misleading and have changed instances of pollination time to pollinator visitation time.

  1. line 244 smallholder farms not farmers

This has been changed accordingly.

  1. line 311 to 319 - I think you need to make this distinction at the start of the manuscript; even "natural pesticides" can have broad spectrum activity and can be harmful to beneficial insects.

We agree that this should be mentioned prior to the discussion and have added the following to line 71: “Although it is chemical pesticides that are primarily harmful to insects, it is true that some biopesticides, i.e., naturally-derived pesticides, can be broad-spectrum and, as a result, damaging to pollinators.”

  1. Conclusions are too long. Reduce by at least 50%. Focus on how your findings relate to your original aims and questions.

We agree that some aspects of the conclusions section were unnecessary and have managed to reduce the conclusions by 40-50%. 

  1. Reducing pesticide use could also help with other aspects of food production including pest control by natural enemies. I do not believe you have mentioned this.

This is an important part that we agree should be addressed. We have added the following to the discussion surrounding the damaging effects of chemical pesticides in line 73: “Not only do non-target species often provide pollination services, but they provide other beneficial services, such as pest control by natural enemies.”.

Reviewer 2 Report

Overall, this is an excellent study, reporting on important results.

One area that could be included is the potential of pollinator diversity leading to greater resilience of this ecosystem function through species complementarity and redundancy. It would further their rationale as to why strategies to foster biodiversity is important, especially in the face of climate change. The potential for biodiversity to contribute to greater resilience balances out the imperative of measuring pollinator diversity against yield. Thus, even if the ‘intensive’ farming resulted in higher coffee yields now, the question remains as to how well such a system would recover if key pollinators were absent or unable to function on their farms.

The discussion is a bit long and moves beyond the results of the study to a more general discourse on “Wildlife – friendly farming”. It’s a great discussion of such farming practices, but does not appear to be directly related to the study, so lines 286 – 330 could be removed to keep the discussion relevant to the present study. At the very least the paragraph describing importance of pollinators and coffee (line 320 – 330) should be moved to the introduction.

Line 136. I’ve only heard of annual crops used in annual rotations and am unfamiliar with the following so could the authors please explain or provide examples of “…perennial crops are rotated annually…”

The last paragraph of the introduction describing the study is written in the present tense. Usually everything is written in past tense because we are reporting on what has been done, so the authors may need to change to the past tense (Lines 118 - 130). 

Author Response

Overall, this is an excellent study, reporting on important results.

One area that could be included is the potential of pollinator diversity leading to greater resilience of this ecosystem function through species complementarity and redundancy. It would further their rationale as to why strategies to foster biodiversity is important, especially in the face of climate change. The potential for biodiversity to contribute to greater resilience balances out the imperative of measuring pollinator diversity against yield. Thus, even if the ‘intensive’ farming resulted in higher coffee yields now, the question remains as to how well such a system would recover if key pollinators were absent or unable to function on their farms.

This is a wonderful suggestion and we believe helps to bolster our discussion surrounding the importance of diversity for long-term ecosystem resilience. We have added brief discussions about species complementarity at line 110 and line 317.

The discussion is a bit long and moves beyond the results of the study to a more general discourse on “Wildlife – friendly farming”. It’s a great discussion of such farming practices, but does not appear to be directly related to the study, so lines 286 – 330 could be removed to keep the discussion relevant to the present study. At the very least the paragraph describing importance of pollinators and coffee (line 320 – 330) should be moved to the introduction.

We agree that the discussion is lengthy and, as a result, have moved line 320-330 (prior to editing) to the introduction in-line with your suggestion. We have kept the wildlife-friendly farming paragraphs due to this being an important aspect of our research and referring to them in this paper is in-line with the holistic approach we are taking in promoting the use of these farming practices.

Line 136. I’ve only heard of annual crops used in annual rotations and am unfamiliar with the following so could the authors please explain or provide examples of “…perennial crops are rotated annually…”

We have come to the conclusion that the term “perennial” is unnecessary and have removed it. We hope that this is no longer a source of confusion for readers.

The last paragraph of the introduction describing the study is written in the present tense. Usually everything is written in past tense because we are reporting on what has been done, so the authors may need to change to the past tense (Lines 118 - 130).

We agree that the description of the study should be in the past tense, but we have kept how we will use our results (i.e. the last two sentences of the final paragraph) in the present tense.